# ErbB4 Is a Potential Key Regulator of the Pathways Activated by NTRK-Fusions in Thyroid Cancer

**Andrey Kechin** [1,2,*], **Viktoriya Borobova** [1,2], **Alexander Kel** [1,3], **Anatoliy Ivanov** [4] **and Maxim Filipenko** [1]

1. Institute of Chemical Biology and Fundamental Medicine, 630090 Novosibirsk, Russia; v.borobova@g.nsu.ru (V.B.); alexander.kel@genexplain.com (A.K.); max@niboch.nsc.ru (M.F.)
2. Faculty of Natural Sciences, Novosibirsk State University, 630090 Novosibirsk, Russia
3. geneXplain GmbH, 38302 Wolfenbüttel, Germany
4. Altai Regional Oncological Dispensary, 656000 Barnaul, Russia; anatolij0199@yandex.ru
* Correspondence: aa_kechin@niboch.nsc.ru

**Featured Application: To our knowledge, this is the first study in which a link between the ErbB4 pathway and NTRK gene fusions has been identified. This association may be a part of the resistance mechanism to Trk-inhibitors and could be used as an indirect molecular marker of such specific thyroid tumor phenotypes. Additionally, it could be helpful for future drug development.**

**Abstract:** NTRK gene fusions are drivers of tumorigenesis events that specific Trk-inhibitors can target. Current knowledge of the downstream pathways activated has been previously limited to the pathways of regulator proteins phosphorylated directly by Trk receptors. Here, we aimed to detect genes whose expression is increased in response to the activation of these pathways. We identified and analyzed differentially expressed genes in thyroid cancer samples with *NTRK1* or *NTRK3* gene fusions, and without any NTRK fusions, versus normal thyroid gland tissues, using data from the Cancer Genome Atlas, the DESeq2 tool, and the Genome Enhancer and geneXplain platforms. Searching for the genes activated only in samples with an NTRK fusion as opposed to those without NTRK fusions, we identified 29 genes involved in nervous system development, including *AUTS2*, *DTNA*, *ERBB4*, *FLRT2*, *FLRT3*, *RPH3A*, and *SCN4A*. We found that genes regulating the expression of the upregulated genes (i.e., upstream regulators) were enriched in the "signaling by *ERBB4*" pathway. *ERBB4* was also one of three genes encoding master regulators whose expression was increased only in samples with an NTRK fusion. Moreover, the algorithm searching for positive feedback loops for gene promoters and transcription factors (a so-called "walking pathways" algorithm) identified the ErbB4 protein as the key master regulator. *ERBB4* upregulation (*p*-value = 0.004) was confirmed in an independent sample of *ETV6-NTRK3*-positive FFPE specimens. Thus, ErbB4 is the potential key regulator of the pathways activated by NTRK gene fusions in thyroid cancer. These results are preliminary and require additional biochemical validation.

**Keywords:** thyroid cancer; NTRK; ErbB4; pathways; gene fusions





## 1. Introduction

Human NTRK genes (*NTRK1*, *NTRK2*, and *NTRK3*) encode three neurotrophic receptors TrkA, TrkB, and TrkC, respectively. Normally, their activation is induced by the binding of specific neurotrophic factors or in response to G-protein coupled receptor signaling in various regulation processes of cell growth, survival, and differentiation [1]. For the nervous system, these processes are manifested mainly in the formation of axons, dendrites, and synapses [2]. In other organs, e.g., thyroid glands, their role is unclear, but their aberrant activation, which happens mainly through gene fusion with other constitutively expressed genes, may lead to the activation of the mitogen-activated protein kinase (MAPK), phosphatidylinositol-3-kinase (PI3K)/AKT, phospholipase C gamma (PLCγ)-Ca$^{2+}$, nuclear

factor-kappa B (NF-κB), and protein kinase C (PKC) pathways, and consequently, tumorigenesis [1]. Tumors developed via this mechanism are sensitive to NTRK-inhibitors, e.g., larotrectenib or entrectinib, which have a high response rate with low toxicity [3]. However, due to the low occurrence of NTRK-fusions (~0.3% in all solid tumors) [4], and relatively recent development of effective methods for NTRK rearrangement detection [5,6], previous studies were mainly focused on the biochemical analysis of activated Trk partner proteins, or the properties of the fused proteins themselves.

Simultaneously, the analysis of differentially expressed genes, with a focus on searching for key master regulators, may help in the identification of new targets for inhibition [7,8]. Additionally, such computational predictions can be confirmed by in vitro studies [9], or even lead to new drug development [10]. Commonly, these analyses include the identification of differentially expressed genes, their prioritization, for example, by the presence in only one of the lists [11], gene network construction, and master regulator identification [12].

Using TCGA data, we investigated genes which were differentially expressed in thyroid tumors containing and not containing NTRK-fusions versus normal thyroid gland data that was recently obtained by Suntsova et al. [13]. As for NTRK-fusion events in other types of tumors, we can identify only a few in the TCGA database (1–4 samples per tumor type), limiting the possibility of performing a tumor-type independent analysis of NTRK-fusions. To our knowledge, this is the first study in which the NTRK-fusion activation mechanism was elucidated through gene expression data analysis.

## 2. Materials and Methods

### 2.1. Datasets

The expression data was downloaded from the Cancer Genome Atlas (TCGA) portal in an un-normalized format (htseq counts), as is suggested in Love et al. [14]. The NTRK-fusion status was determined using the Tumor Fusion Gene Data portal (https://tumorfusions.org/, accessed on 7 December 2021) [15]. Among the samples with NTRK-fusions, we identified five with *NTRK1-* (two cases were with *IRF2BP2-NTRK1*) and six with *NTRK3*-fusions (five cases were with *ETV6-NTRK3*, other sample information can be found at GitHub: https://github.com/aakechin/erbb4_ntrk_fusions/, accessed on 8 December 2021). The number of tumor samples without NTRK rearrangements was 570. The gene expression values for normal thyroid gland tissues (six samples) were downloaded from the Gene Expression Omnibus database (GSE120795) [13].

### 2.2. Data Analysis

#### 2.2.1. Identification of Differentially Expressed Genes (DEGs)

The data preprocessing, normalization, identification of differentially expressed genes, and visualization were carried out in R using the DESeq2 [14] and ggplot2 [16] libraries. The R and bash scripts, and the geneXplain pipeline used in this work can be found at GitHub, project: https://github.com/aakechin/erbb4_ntrk_fusions/ (accessed on 8 December 2021). In GitHub, the file Workflow_NTRK.dml provides the XML code of the workflow to analyze a gene set and to predict master regulators. To execute the workflow, the file should be uploaded to the geneXplain platform: https://platform.genexplain.com (accessed on 8 December 2021). Licenses for the databases TRANSFAC, TRANSPATH, and HumanPSD are required to execute the full workflow. For the DEG analysis, a parametric fit type was used. The gene expression values were compared for tumor samples without NTRK-fusion versus normal tissue, samples with an *NTRK1* gene fusion versus normal tissue, and samples with an *NTRK3* gene fusion versus normal tissue. There were no thyroid cancer samples with an *NTRK2* fusion. The comparison results were saved as CSV files and imported into the geneXplain platform (https://genexplain.com/genexplain-platform/, accessed on 8 December 2021). A Venn diagram analysis of the geneXplain platform was used to compare lists of DEG genes.

### 2.2.2. Gene Set Enrichment Analysis

Gene set enrichment analysis was performed in the geneXplain platform using the "full gene ontology classification" and "reactome pathways (74)" databases.

### 2.2.3. Master Regulator Analysis and Upstream Analysis

To search for master regulators for the differentially expressed genes, a "regulator search" analysis was applied (using the GeneWays database [17] and a maximal radius of 2 for the upstream search). The analysis of key master regulators was done with the help of the software Genome Enhancer (my-genome-enhancer.com, accessed on 8 December 2021) [18]. Genome Enhancer implements the "upstream analysis" method [19,20] that provides a casual interpretation of the expression regulation of the genes analyzed in our study. Here, the upstream analysis is empowered by the search for positive-feedback loops [21], the so-called algorithm of "walking pathways" [18], that makes it more focused compared to the "regulator search". This approach comprises two major steps: (1) analyzing the promoters and enhancers of differentially expressed genes for the transcription factors (TFs) involved in their regulation and, thus, important for the process under study; (2) reconstructing the signaling pathways that activate these TFs, and identifying master regulators at the top of such pathways. For the first step, the database TRANSFAC® [22] (release 2021.1) was employed, together with the TF binding site identification algorithms Match [15], F-Match [16], and Composite Module Analyst (CMA) [17]. The Match tool searches for TF binding sites in the promoters of input genes using the full collections of 7889 vertebrate position weight matrices (PWMs) from the TRANSFAC database, applying PWM cut-offs via "min_SUM" which provides the minimum of the sum of false positive and false negative rates in site prediction. The F-Match tool identifies PWMs for transcription factors whose sites are statistically significantly enriched in the promoters of the input gene set in comparison with the background set of promoters (here, a set of 1500 housekeeping genes was used as the background set). The CMA tool searches for composite modules—combinations of the enriched PWMs whose sites are co-localized in the promoters of the studied genes and build compact clusters. Such composite modules often serve as functional units for the regulation of gene expression in specific conditions (e.g., in tumor cells). The second step involves the signal transduction database TRANSPATH® [23], and special graph search algorithms [21] implemented in the software Genome Enhancer. The difference was considered statistically significant if the *p*-value was less than 0.05 after a multiple testing correction (the Bonferroni test or Benjamini-Hochberg procedure were used to control the false discovery rate).

### 2.3. Experimental Validation

To confirm the observed difference in *ERBB4* expression between samples containing and not containing NTRK-fusions, we screened 146 *BRAF* V600E-negative *ETV6-NTRK3*-positive papillary thyroid cancer FFPE blocks obtained from Altai Regional Oncological Dispensary. RNA was isolated from three 5 μm thick sections using the RNeasy High Pure FFPET RNA Isolation kit (Roche Life Science, Penzberg, Germany). NTRK-fusions were detected with ETV6-NTRK3-specific primers and probes, while *ERBB4* and *TBP* expression were analyzed via quantitative RT-PCR with specific primers and probes (Table 1). The successive reverse transcription and PCR reactions were performed in 25 μL containing TaqPath 1-Step Multiplex Master Mix (Thermo Fisher Scientific, Waltham, MA, USA), 450 nM of primers, and 150 nM of probes. To construct the calibration curve, sample RNA analysis was accompanied by the amplification of standard synthetic plasmids involving target sequences in five concentrations in duplicate. For normalization, the TBP expression was used.

**Table 1.** Oligonucleotide primers and fluorescently labeled probes used for the *ERBB4* expression estimation in the FFPE blocks with and without NTRK gene fusions.

| Genes | Primers and Probes |
|---|---|
| *TBP* | TBP-R2 5′-CACTGTGGATACAATATTTTGCAG-3′<br>TBP-U2 5′-CACTCCTGCCACGCCAGCT-3′<br>TBP-PC2R 5′-ROX-CGGAGAGTTCTGGGATTGTACCG-BHQ2-3′ |
| *ERBB4* | ERBB4-U2 5′-CCAGATCGGGAGTGCCACC-3′<br>ERBB4-R2 5′-ATGACTAGTGGGACCGTTACAC-3′<br>ERBB4-PF2 5′-FAM-TGCCATCCAAACTGCACCCAAGG-BHQ1-3′ |
| *ETV6-NTRK3* | ETV6-NTRK3-R 5′-CACCCAGTTCTCGCTTCAGC-3′<br>ETV6-NTRK3-U 5′-GAGCACGCCATGCCCATTG-3′<br>ETV6-NTRK3-PH<br>5′-HEX-CAGCACATTAAGAGGAGAGACATCG-BHQ1-3′ |

## 3. Results

### 3.1. Differentially Expressed Genes

To find genes that are activated in response to NTRK-fusion formation, we analyzed the RNA-seq data and identified DEGs in the following comparisons: (1) thyroid cancers without NTRK-fusions versus normal thyroid gland tissues; (2) thyroid cancer with *NTRK1*-fusions versus normal thyroid gland tissues; (3) thyroid cancer with *NTRK3*-fusions versus normal thyroid gland tissues. In order to select a high number of upregulated genes in response to NTRK-fusion, we identified genes with a log2FoldChange $\geq 1$ in any of these three comparisons and used them for further analysis. The *p*-values were not used for gene filtering due to the low number of samples with NTRK-fusion, which could lead to the unreliable estimation of statistical significance and missing identification of key upregulated genes. We compared the three obtained lists of genes using the Venn diagram method (Figure 1A). About 90% of upregulated and downregulated genes were common for all three comparisons (Figure 1B). We identified 8809 upregulated genes common in all three subtypes—*NTRK1*-fusions, *NTRK3*-fusions, and no NTRK fusion, comparing tumor tissue vs. normal tissue. Gene set enrichment analysis (GSEA) of the GO and pathways for these common genes identified among the most enriched categories—"adaptive immune response", "ER and Golgi membranes", "degradation of purine ribonucleotides", "ubiquitination", "gluconeogenesis", and "the citric acid cycle (TCA) and respiratory electron transport." Thus, our analysis demonstrated that cancerogenic mechanisms are greatly shared between all subtypes of thyroid tumors, which suggests the potential for common therapeutic opportunities for all thyroid tumors.

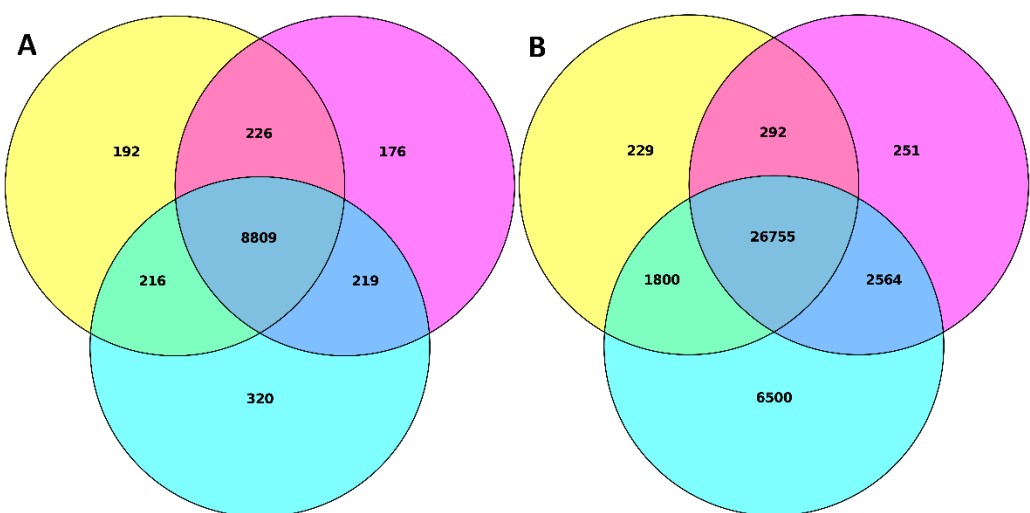

**Figure 1.** The comparison of upregulated (**A**) and downregulated (**B**) gene lists for samples with *NTRK1*- or *NTRK3*-fusion, and without any NTRK gene fusions.

### 3.2. Genes Upregulated Only in Samples with NTRK1 and NTRK3 Gene Fusions

For our study, the genes activated in both *NTRK1* and *NTRK3* sample groups were of the most interest (226 genes). GSEA showed that many of these genes were involved in nervous system development processes, e.g., axonogenesis, synapse assembly, neuron differentiation, and neuron migration (Figure 2).

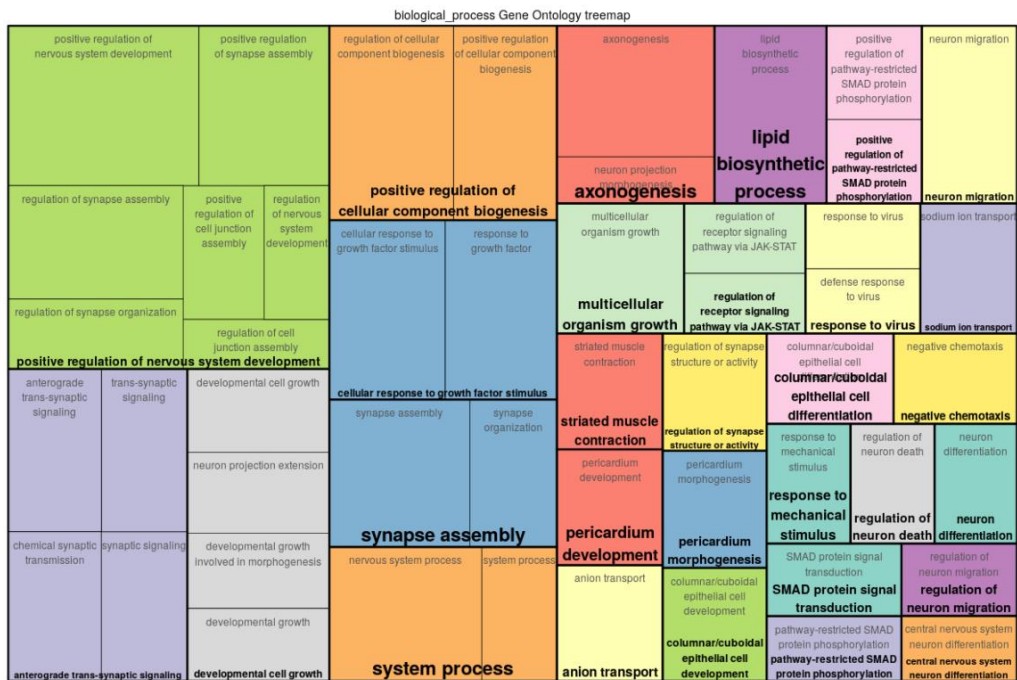

**Figure 2.** Tree map of biological processes found by GSEA of the upregulated genes in the *NTRK1* and *NTRK3* tumors.

### 3.3. Regulators for the Genes Upregulated in Samples with NTRK1 and NTRK3 Gene Fusions

To determine which proteins were engaged in activating the gene set observed, we performed an upstream search for the regulators of these genes, using the GeneWays database of the human gene regulatory network, employing the key-node search algorithm "regulator search" of the geneXplain platform (Supplementary Table S1). Among the top 10 regulators found, we identified: SRC, STAT5A, LCK, VAV1, STAT3, STAT1, MYO15A, EGF, CBL, and FYN. In addition, the log2FoldChange values for all discovered regulatory genes showed that three of them were equally upregulated in the samples with *NTRK1* and *NTRK3* gene fusions, but not in the no-NTRK samples: *ERBB4*, *BMX*, and *BMPR1B* (Figure 3).

GSEA for all regulators using the "reactome pathways" database identified two small pathways, "signaling by EGFR in cancer" (false discovery rate, FDR = 0.016) and "signaling by ERBB4" (FDR = 0.038) (Supplementary Table S2). We compared the normalized expression value distributions for the genes that belong to these two pathways of signaling by EGFR and ERBB4 in normal thyroid gland tissues, in thyroid cancers without an NTRK gene fusion, and in thyroid cancers with *NTRK1-* and *NTRK3*-fusions (Figure 4). The median normalized values were higher for all genes for samples with *NTRK1-* or *NTRK3*-fusions, with the highest difference for *ERBB4*, *BMX*, and *EPOR*.

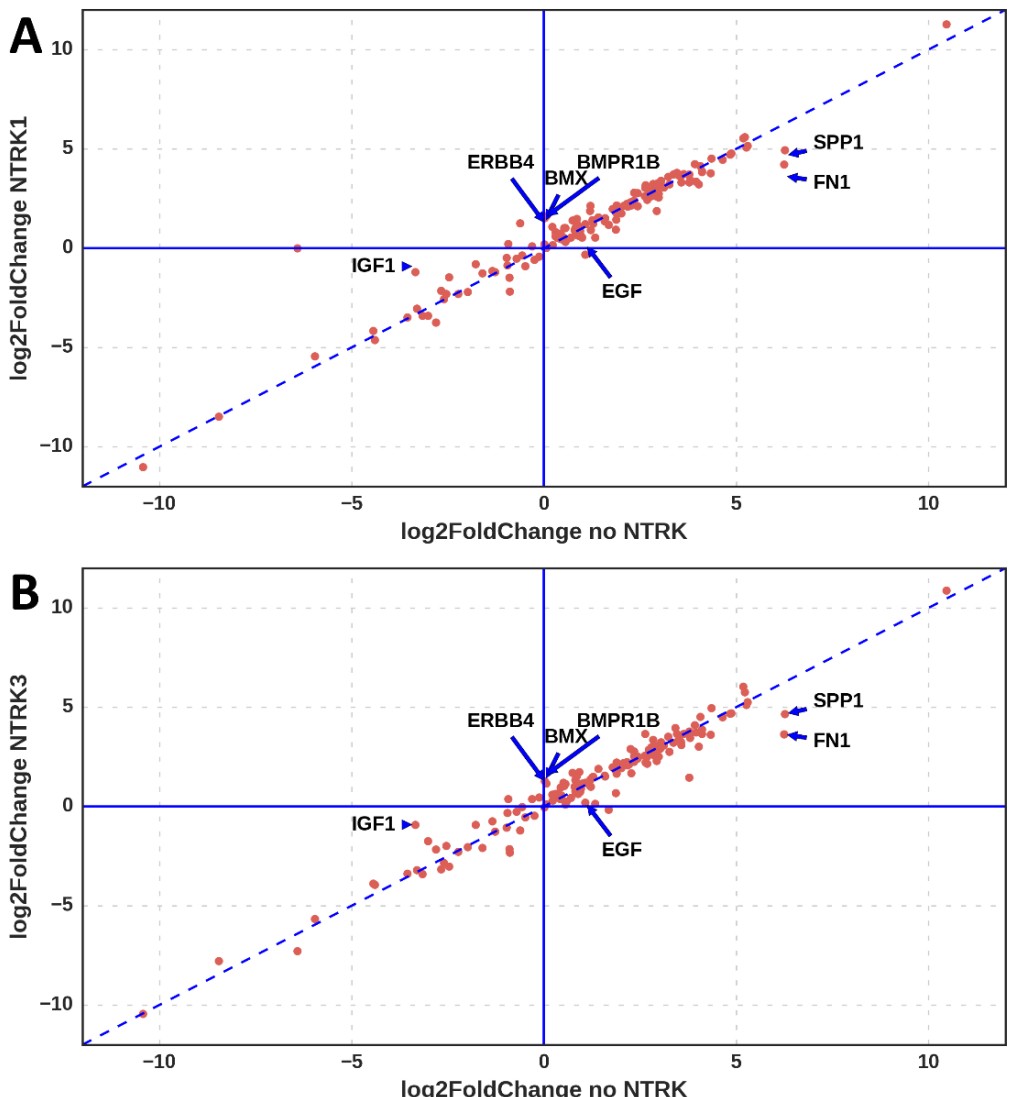

**Figure 3.** The log2 transformed foldchange values for genes encoding the upstream regulators identified. Samples with *NTRK1* gene fusions versus samples without NTRK fusions (**A**) and samples with *NTRK3* fusions versus samples without NTRK fusions (**B**). Genes with maximally different log2FoldChange values are annotated with arrows and gene symbols.

*3.4. Regulators of Upregulated Genes Associated with the Nervous System*

Returning to the fact that genes associated with nervous system development were upregulated only in the thyroid cancer samples with *NTRK1* or *NTRK3* gene fusions (Figure 5), we specifically searched for upstream regulators for these genes. In total, we identified 29 upregulated genes that were related to the following GO categories associated with the nervous system (e.g., "axon", "nervous system process", "synapse assembly", "axonogenesis") (Supplementary Table S3). Among them, the highest gene expression difference between the samples with and without NTRK-fusions was for the genes: *AUTS2*, *DTNA*, *ERBB4*, *FLRT2*, *FLRT3*, *RPH3A*, and *SCN4A*.

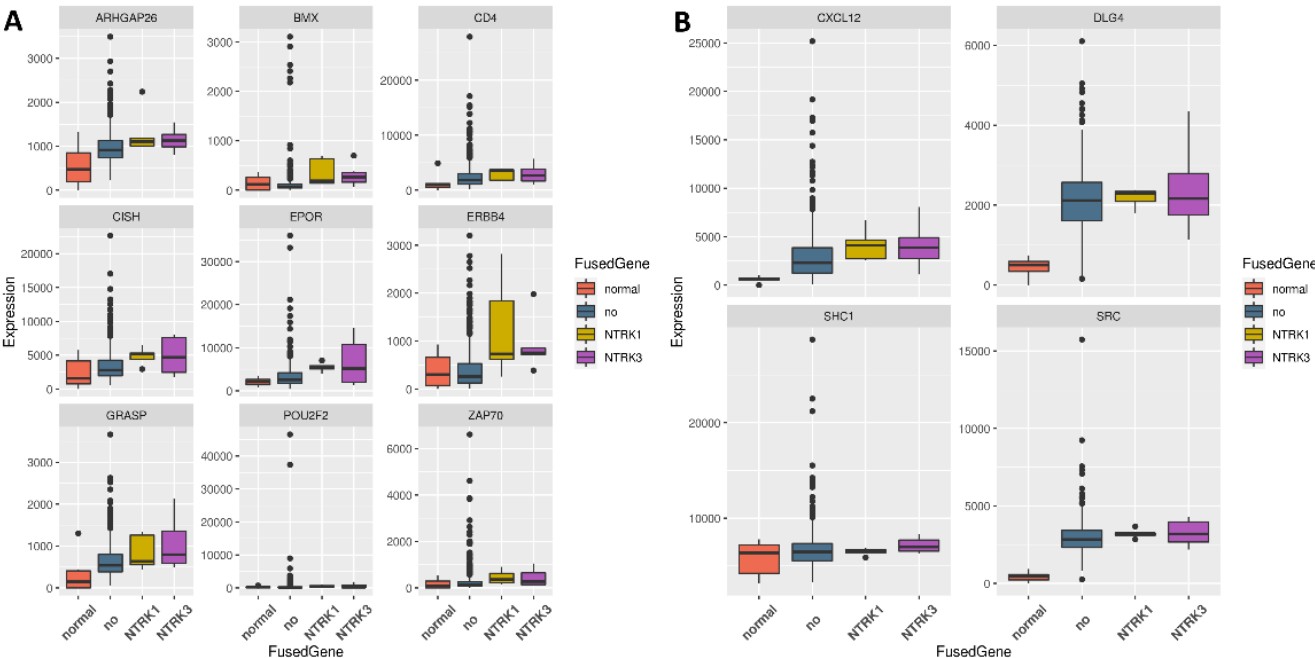

**Figure 4.** The comparison of the normalized gene expression value distributions for the genes regulated by *EGFR* (**A**) and *ERBB4* (**B**) in normal thyroid gland tissues (first boxes), thyroid cancer without NTRK gene fusion (second boxes), with *NTRK1* (third boxes) and *NTRK3* (fourth boxes) gene fusions.

We suggest that the activation of these genes is a consequence of NTRK-fusion occurring in thyroid tumor tissue and the subsequent rewiring of the key signaling pathways in these tumors towards the further activation of their growth. Such potential rewiring in the network can be identified with the help of the search for positive feedback loops, "walking pathways" [18], that is implemented in the Genome Enhancer tool. Therefore, we applied the Genome Enhancer tool to analyze the 29 genes identified to find key master regulators that may enhance their expression in the studied tumor types. During the first step, Genome Enhancer analyzed the promoters of the 29 upregulated genes, applying the F-Match [19] and CMA [24] tools using the TRANSFAC motif library [25]. As a result, clusters of binding sites were revealed for the following transcription factors: NFkB, Oct-1, Egr2, and KLF4 in the promoters of these up-regulated genes (Supplementary Figure S1), explaining the details of the molecular mechanism of their transcriptional control in tumors of the studied types. Next, Genome Enhancer, with the help of the algorithm "walking pathways" [18], used the database TRANSPATH [23] and performed a search in the signal transduction network upstream of the revealed transcription factors, and revealed four key master regulators (encoded by the genes *ERBB4*, *DKK1*, *NOG*, and *ARHGEF7*). ErbB4 was determined as the top key master regulator. Finally, we reconstructed the potential regulatory network connecting activated ErbB4 and a few other key master-regulators molecules with the identified upregulated nervous system development genes in the thyroid tumors carrying *NTRK1*- and *NTRK3*-fusions (Figure 6).

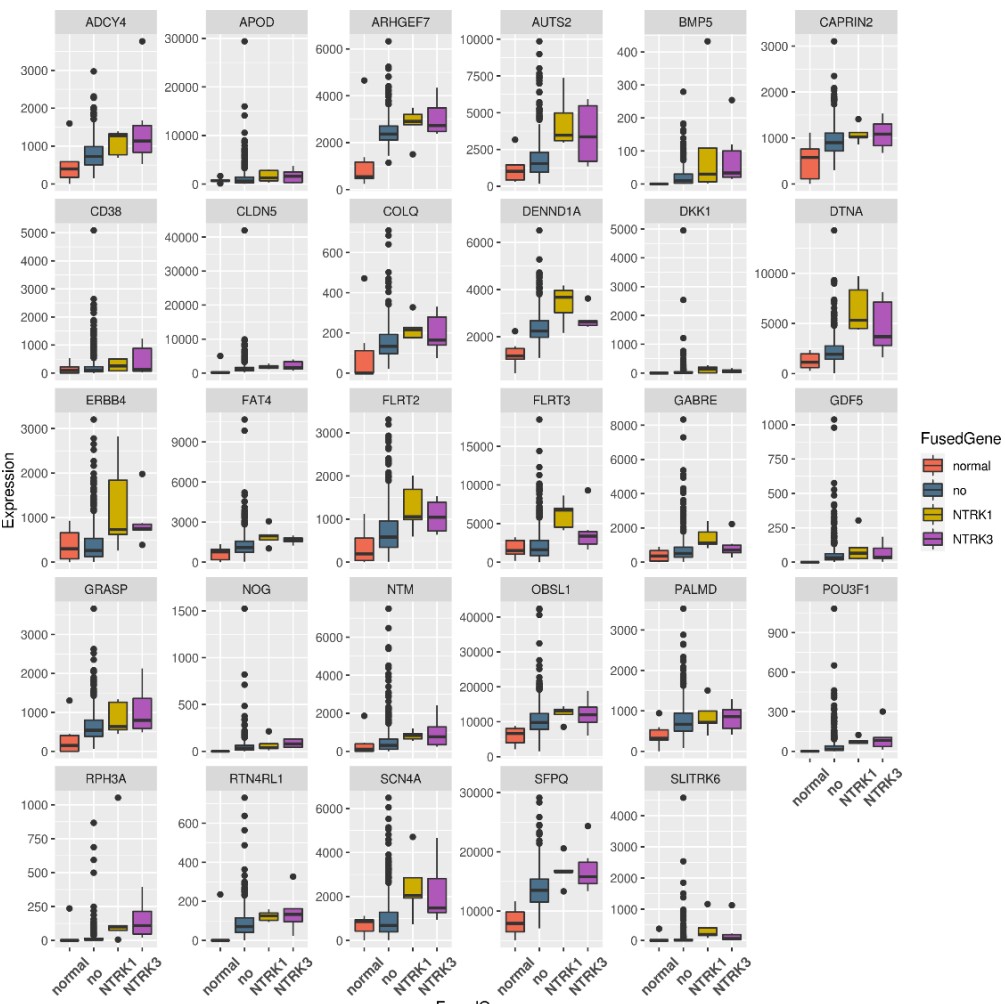

**Figure 5.** The comparison of the normalized gene expression value distributions for the genes associated with nervous system development in normal thyroid gland tissues (first boxes), thyroid cancer without NTRK gene fusion (second boxes), with *NTRK1* (third boxes) and *NTRK3* (fourth boxes) gene fusions.

Using the STRING database, we constructed an interaction network for the top upregulated genes and associated master regulators identified in our analysis (Figure 7). For proteins encoded by *ERBB4*, *EGFR*, *BMX*, *DKK1*, *NOG*, *ARHGEF7*, and *NTRK1*, various interactions were documented in the STRING database—experimentally determined physical interactions and co-expression of their homologs in other organisms, or relations between genes found by text mining of scientific publications. Some of these interactions may play an essential role during thyroid cancer development.

### 3.5. ERBB4 Was Significantly Upregulated in Independent Samples with NTRK3 Fusion

Utilizing RT-PCR, we selected five RNA samples from papillary *BRAF* V600E negative thyroid cancer FFPE blocks containing the most frequent *ETV6-NTRK3* fusion and 10 RNA samples without *ETV6-NTRK3* fusion. The *ERBB4* expression measured by qRT-PCR was significantly higher in samples with the *NTRK3* rearrangement ($p$-value = 0.004 in Mann–Whitney two-sided test) (Figure 8).

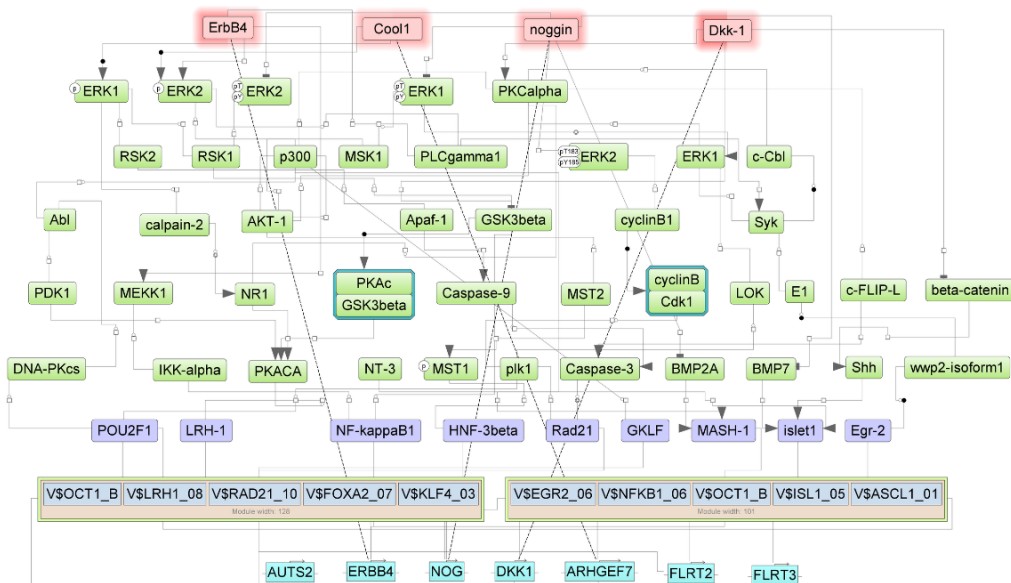

**Figure 6.** The potential signal transduction and gene regulatory network acting in the thyroid tumors, with *NTRK1* and *NTRK3* gene fusions activating nervous system development genes. The revealed activated key master-regulator proteins ErbB4, Cool1, Noggin, and Dkk-1 (pink nodes on the top) through the intracellular signal transduction network (green nodes) activate the transcription factors (blue nodes) that, in turn, upregulate the genes of nervous system development and axon growth (light blue nodes in the bottom of the figure). The dotted lines indicate a transcription-translation link from some of these genes (*ERBB4*, *NOG*, *DKK1*, *ARHGEF7*) to the proteins encoded by these genes (positive feedback loops). The red shadows around key master regulators correspond to the expression fold changes in their genes in the tumors carrying *NTRK1*- (right half of the shadow) and *NTRK3*- (left half of the shadow) fusions, respectively.

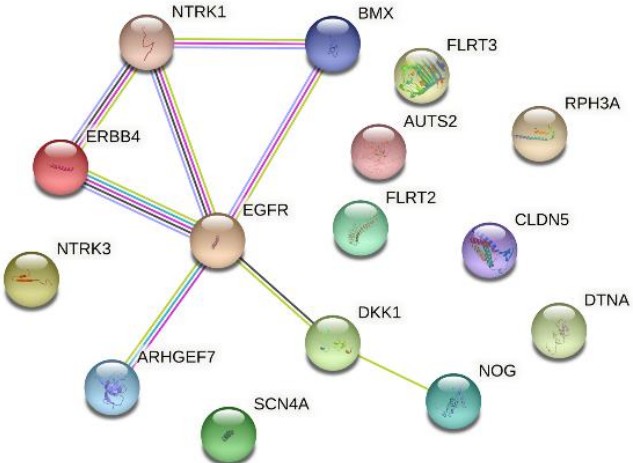

**Figure 7.** STRING network analysis of genes suggested in this study as participants in signal transduction from TrkA and TrkC activated by gene fusion formation.

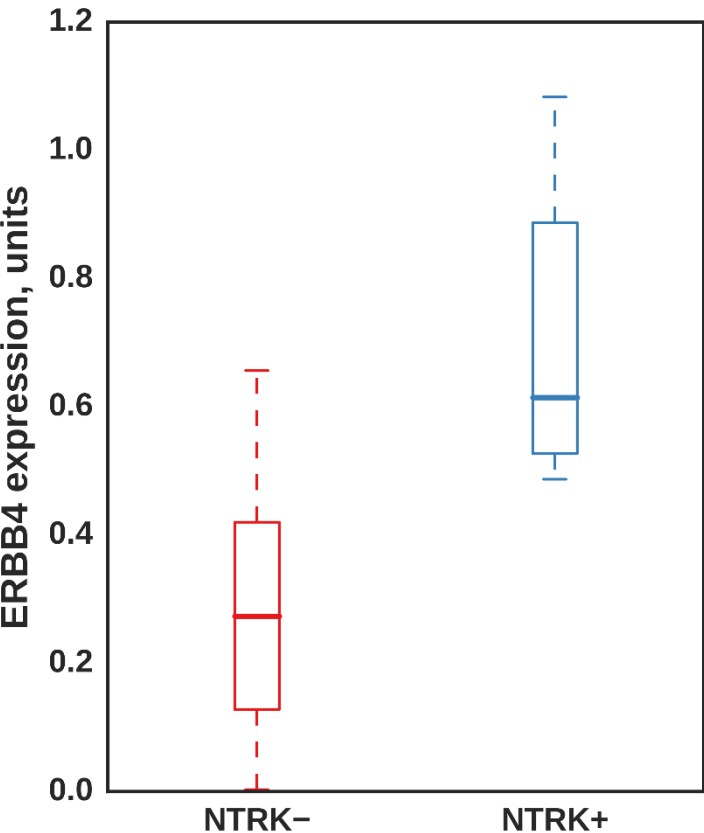

**Figure 8.** *ERBB4* gene expression values in samples with (NTRK+) and without (NTRK−) NTRK gene rearrangements.

## 4. Discussion

In the current study, analyzing TCGA gene expression data, we suggested genes that may participate in the tumorigenesis induced by NTRK gene fusions: *ERBB4*, *EGFR*, *BMX*, *BMPR1B*, *AUTS2*, *DTNA*, *ERBB4*, *FLRT2*, *FLRT3*, *RPH3A*, *DKK1*, *NOG*, *ARHGEF7*, and *SCN4A*. All but *BMX* and *ERBB4* were statistically significantly upregulated in thyroid cancers, both with *NTRK1* or *NTRK3* gene fusions compared to normal thyroid gland tissue. However, *ERBB4* was statistically significantly upregulated only when comparing both NTRK sample subtypes taken together versus normal tissue samples (log2FC = 4.7; *p*-value = 0.009). Based on the results of the comparison of differential gene expression in different tumor subtypes, pathway analysis, STRING analysis, and master regulator searches, we focused most of our attention on seven of those genes that may be considered as the more likely participants of malignancy pathways activated by TrkA or TrkC: *ERBB4*, *EGFR*, *BMX*, *AUTS2*, *DKK1*, *ARHGEF7*, and *FLRT3* (Table 2).

**Table 2.** The comparison of genes suggested as participants of pathways activated by fused Trk. DEG is a differentially expressed gene.

| Gene | DEG with *p*-Value < 0.05 | Identified as Regulator | Expression More than in Samples without NTRK-Fusions | Interactions with Trk Found in STRING | Known Engagement in Malignancy |
|---|---|---|---|---|---|
| *ERBB4* | yes/no | yes | yes | yes | yes |
| *EGFR* | yes | yes | no | yes | yes |
| *BMX* | no | yes | yes/no | yes | yes |
| *AUTS2* | yes | no | yes | no | yes |
| *DKK1* | yes | yes | yes | yes | yes |
| *ARHGEF7* | yes | yes | yes | yes | yes |
| *FLRT3* | yes | no | yes | no | no |

In this work, in addition to the standard methods of pathway mapping and STRING analysis, we applied one of the most advanced approaches to the bioinformatic analysis of gene regulatory networks—an upstream analysis and master regulator search implemented in the set of tools provided by the geneXplain platform and the Genome Enhancer tool. The upstream analysis approach tries to reconstruct the causal interactions in the signal transduction and gene regulatory networks that are heavily rewired in the tumor cells, and cannot be properly mapped to canonical pathways that are acting in a non-tumor cellular context. To do such causal reconstruction, the most comprehensive collection of TF binding site motifs provided by the TRANSFAC database was necessary to minimize the chances of missing an important TF-DNA interaction playing a vital role in the tumorigenic process. We also needed to understand which particular and rather novel combination of transcription factors is responsible for such very specific gene regulation as was observed in the studied subtypes of tumors. The consequent use of the Match, F-Match and CMA tools provided us this possibility, which is hardly possible with other tools, such as "MatrixCatch" [26], which searches only known composite elements but cannot find new combinations, and "PC-TraFF" [27,28] which searches for TF site pairs in the whole promoter length but not more complex combinations of co-localized TF sites. At the last step of the causal reconstruction of the rewired networks, we searched for the most probable regulators and key master-regulators. This was done with the use of the GeneWays and TRANSPATH databases that gave us, on one side, a very broad scope of interactions (provided by the text-mining based database GeneWays) and, on the other, highly accurate information about the details of signal transduction reactions (provided by the manually curated and highly comprehensive database TRANSPATH). This combination of two databases helped us to achieve the most reliable results of our analysis.

The *ERBB4* gene is a member of the epidermal growth factor receptor family and plays a vital role in normal tissue development, especially in the nervous system [29–31]. For some tumors, its expression has prognostic value [32], and for some tumors, it is also considered a prospective therapeutic target [33,34]. The *EGFR* gene is another member of the EGFR family whose role in tumorigenesis is well known, and many small molecule inhibitors and monoclonal antibodies have been developed to treat non-small-cell lung cancer, pancreatic cancer, breast cancer, and colon cancer [35]. However, EGFR also has a significant kinase-independent pro-survival function, whose mechanism is mainly unknown but may lead to anti-EGFR resistance [36]. Another known example of anti-EGFR resistance is the activation of BDNF/TrkB signaling [37] that links Trk-receptor functionally with EGFR.

*BMX* is a gene encoding a nonreceptor tyrosine kinase [38] for which participation in myocardial hypertrophy [39], inflammation [40], keratinocyte proliferation [41], and castration-resistant prostate cancer development [42] is known. Moreover, a recent study showed that a BMX-ARHGAP fusion protein might activate the JAK/STAT3 signaling pathway [43], which contributes to gastric cancer tumorigenicity.

*FLRT3* is a gene of the FLRT gene family participating in somite development [44]. However, its role in endothelial cell survival, migration, and tube formation after the vascular endothelial growth factor (VEGF) binds to VEGF receptor (VEGFR) was recently shown [45]. Furthermore, another gene, *AUTS2*, in addition to its essential role in synapse formation [46,47], has been shown to be a master regulator of genes for the development of the nervous system [48]. Moreover, for the Drosophila melanogaster homolog of *AUTS2*, Tay Bridge, negative regulation of EGFR-signaling has been shown [49] that may shed light on the link between ERBB4 and AUTS2 identified in our data.

Interestingly, we can confirm the possible use of some of the identified master-regulator proteins as diagnostic and prognostic biomarkers by comparison with recently published independent proteomics data of papillary thyroid carcinoma samples [50]. Among more than 2500 proteins identified in that study as specifically detected or overexpressed in the samples of thyroid tumors, two top proteins matched the master regulators identified in our work: Dkk-1 (DKK1), and Cool-1 (ARHGEF7). The protein Dkk-1 (Dickkopf homolog 1) is

an LDL receptor ligand that acts in the canonical Wnt signaling pathway. It is upregulated in Alzheimer's disease, nerve degeneration, ankylosing spondylitis, osteoporosis, and lung and several other cancers. It was earlier shown that Dkk-1 overexpression triggers an activation of the Wnt pathway that contributes to resistance to the antitumor drug Ibrutinib, targeting ErbB4 [34]. The intriguing role of the Cool-1 (ARHGEF7) protein in cancerogenesis was described earlier [51]. Cool-1 is a Rho guanine nucleotide exchange factor 7 that plays a role in transmembrane receptor protein tyrosine kinase signaling, synaptogenesis, and cytoskeleton organization, regulating cell shape and dendrite development. The whole-genome sequencing of metastatic tumors recently revealed the *ARHGEF7* gene as playing a central role in metastasis evolution [52].

The identified potential participants and regulators of the pathways activated by the NTRK gene fusion formation may partially explain the resistance mechanism to Trk-inhibitors and propose new targets for overcoming this resistance. The results should be considered preliminary because they include only limited confirmatory biochemical experiments. However, these data should stimulate researchers to search for new physical and biochemical interactions between the proteins identified. Another way to confirm these results is an extension of the sample size with NTRK fusions. Additionally, the latter is expected to be carried out as the number of patients with this gene rearrangement increases due to new diagnostic tools [2].

Our focus on thyroid cancer samples in this study was due to the most frequent cases of NTRK fusions among different tumor types in the TCGA database (11 samples versus 4 samples of brain tumor—the next most frequent tumor type with NTRK gene fusions). The combination of all NTRK fusion-positive samples with different tumor types could increase the statistical power of this analysis. However, an accurate DEG analysis versus normal tissue specimens is necessary to overcome the high transcriptomic difference between tissue types.

Thus, this new data regarding the possible molecular mechanisms linking Trk activation with ErbB4 is preliminary, but may have a high impact on further studies in this scientific field.

## 5. Conclusions

Thus, using several modern bioinformatic approaches, we managed to suggest a new role for ErbB4 in pathway activation due to NTRK gene fusions in thyroid cancer. This new data about the possible molecular mechanisms linking Trk activation with ErbB4 is preliminary, but may have a high impact on further studies in this scientific field.

**Supplementary Materials:** The following supporting information can be downloaded at: https://www.mdpi.com/article/10.3390/app12052506/s1, Table S1: The regulator genes enriched for the genes upregulated in both the *NTRK1* and *NTRK3* groups but not in the no NTRK group; Table S2: Reactome pathways enriched by the regulators identified for the *NTRK1* and *NTRK3* sample groups; Table S3: GO categories associated with the nervous system, in which 29 upregulated genes were enriched. Figure S1: Composite modules (clusters) of binding sites for synergistically acting transcription factors (NFkB, Oct-1, Egr2, KLF4, HNF3B/FOXA2, and others) identified in the promoters of up-regulated genes involved in nervous system development in thyroid tumors carrying NTRK1 and NTRK3 gene fusions. (A) The structure of two composite modules identified in the promoters. Each module consists of 5 TRANSFAC position weight matrices (PWMs) with their optimized cut-offs and maximal number of top scoring sites (N), and the optimized module width. The high quality of the model is charac-terized by the highly statistically significant level of the Wilcoxon test and the AUC value = 0.97. (B) Two histograms that characterize the promoter score distributions in the promoters of up-regulated genes (red bars) and in promoters of housekeeping genes (blue bars). There is a clear separation of these two distributions. (C) An example of the clusters of TF sites that belong to the composite modules identified by the algorithm in the promoter of the ERBB4 gene. The track "Gene" shows the beginning of the first exon of the gene, the dotted vertical line shows the position of the TSS. The track "yes track" represents the identified TF binding sites of the composite modules. The track "all meta clusters" shows the positions of the experimentally

identified binding regions of various transcription factors from publicly available ChIP-seq data (from GTRD database []). One can see the co-localization of several predicted TF binding sites and respec-tive experimentally identified TF binding regions for TF from the same families (Egr-2—EGR3; NF-kappaB1—RELA; KLF4—KLF4/1/5; HNF-3beta—FOXA2).

**Author Contributions:** Conceptualization, M.F. and A.K. (Alexander Kel); methodology, A.K. (Andrey Kechin); software, A.K. (Andrey Kechin) and V.B.; validation, A.I. and M.F.; formal analysis, A.K. (Andrey Kechin); investigation, A.K. (Andrey Kechin) and V.B.; resources, A.I. and V.B.; data curation, V.B.; writing—original draft preparation, A.K. (Andrey Kechin); writing—review and editing, M.F., A.K. (Alexander Kel) and A.K. (Andrey Kechin), visualization, A.K. (Andrey Kechin) and A.K. (Alexander Kel); supervision, A.K. (Alexander Kel); project administration, M.F.; funding acquisition, M.F. All authors have read and agreed to the published version of the manuscript.

**Funding:** This research was funded by the Russian Scientific Foundation, grant number 20-15-00418, "Investigation of molecular mechanisms involved in development of resistance of tumor cells with chimeric NTRK proteins to tyrosine kinase inhibitors in vitro".

**Institutional Review Board Statement:** This study was conducted in accordance with the Declaration of Helsinki, and approved by the Institutional Review Board of the Institute of Chemical Biology and Fundamental Medicine (protocol code 12 from 17 March 2021).

**Informed Consent Statement:** Informed consent was obtained from all subjects involved in the study.

**Data Availability Statement:** The data that support the findings of this study are available in the TCGA repository, the GitHub repository, and Supplementary Materials.

**Conflicts of Interest:** The authors declare no conflict of interest.

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
