# Peer review of "ErbB4 Is a Potential Key Regulator of the Pathways Activated by NTRK-Fusions in Thyroid Cancer"

_applsci, doi:10.3390/app12052506_

Round 1

Reviewer 1 Report

ErbB4 is a potential key regulator of the pathways activated by NTRK-fusions in thyroid cancer

The manuscript by Kechin et .al. performed a systematic bioinformatics analysis to study NTRK fusions in thyroid cancer by collecting the corresponding datasets from different databases.   Although the idea is quite interesting, there are several major problems related to the bioinformatics pipelines, the use of different pathway databases, the introduction section, the MM section, the results section, and the discussion/conclusion sections. 

Regarding the Introduction: The introduction is extremely short and does not provide adequate background information. There are no clear explanations or references to previous biological or bioinformatics studies.

Regarding the MM section: I n my opinion, this is the most problematic part of the manuscript. 

  • First, I wonder why the authors use different pathway databases for their analysis, although according to the documentation of the -genexplain platform- all their analyses can be performed with only one of these databases. It needs to be explained why these different pathway databases were preferred and what is the difference between these pathway databases (Reactome, Transpath, Genways).
  • According to the -genexplain platform- documentation, the underlying algorithm of "regulator search" and "upstream analysis" is the same. Both methods perform promoter analysis, find TFBSs and then identify master regulators.  It must be explained why the authors distinguish between them
  • Transfac database: which version and which PWM library with which cutoffs: this information is completely missing
  • The authors must clearly mention that the -genexplain-platform-, the -genome-enhancer-platform- and the Transfac- and Transpath-databases are commercial. Therefore, the scripts provided in GitHub are only partially applicable. Otherwise, the link from GitHub is missunderstandable
  • In general, the section on data analysis needs to be divided into several subsections. This is because, in my view, the authors should distinguish between master regulator analysis and gene set enrichment analysis. Also, it needs to be explained what a master regulator is with its underlying detection algorithm
  • I wonder what the "Benjamini-Hochberg procedure for calculating false discovery rate" means. I know the Benjamini-Hochberg procedure used to control the false discovery rate. But I don't know how to calculate FDR based on BH procedure. Please explain it.
  • Another problem is the use of the CMA method, which was published in 2006. First, the full name must be written before the abbreviation. But the authors need to explain why this approach is preferred in this study, although members of the genexplain platform have published new studies for the same purpose of "Composite Module Analysis". The studies are
  •  "MatrixCatch" by A. Kel and E. Wingender (published 2013, Bmc Bioinformatics), and MatrixCatch is also integrated with genexplain.
  • "PC-TraFF" and its update by E. Wingender (published 2015 Bmc Bioinformatics and 2018 Frontiers), 
  • In this section, the explanation for the " walking pathways" must also be added.

Regarding the results section: 

  • In my opinion, the sentence in lines 118-119 is misleading: "...To select the most activated genes in response to NTRK fusion, we took genes with log2FoldChange≥1 ..." What does "most activated genes" mean: i) a very high number or ii) a very strong activation? If the authors wanted to study a very high number of genes, I could understand the use of a log2FC>=1, but for the most activated genes this threshold is not valid. This point needs to be clarified.
  • It is not clear to me how these numbers sum up in Figure 1 (B). Is there a total of 38391 down-regulated DEGs discovered by the authors, even though the Ensembl database currently provides information on 20465 vertebrate coding genes? This point needs to be explained.
  • In line 180, the authors mention enriched GO categories, which are listed in Supplementary Table 3. But none of these GO categories has an FDR value of less than 0.05. This threshold is defined by the authors for significance in section MM. It must be explained why these GO terms should be accepted as enriched even though their FDR values are not significant
  • In lines 192 to 194, the authors mention F-Match, an important tool that must be explained in the MM section. Further, the same lines say "... using the TRANSFAC motif library": which version of Transfac, which motif library. I think the authors are talking about a PWM library. If yes, it must be written clearly which PWM library with how many PWMs and with which threshold (minFN, minFP or minSUM or something else).

Regarding the discussion section: there is no relevant information about bioinformatics analysis. Why these databases and tools were selected, what are their advantages and disadvantages. The comparison or ignorance of other tools, as mentioned above. The discussion needs to be expanded in terms of bioinformatics analysis of the manuscript.

Regarding the conclusion section: To meet the requirements for good academic work, a relevant conclusion is required. Please provide a proper conclusion

Author Response

  1. The manuscript by Kechin et .al. performed a systematic bioinformatics analysis to study NTRK fusions in thyroid cancer by collecting the corresponding datasets from different databases.   Although the idea is quite interesting, there are several major problems related to the bioinformatics pipelines, the use of different pathway databases, the introduction section, the MM section, the results section, and the discussion/conclusion sections. 

Thanks for the comments and suggestions for improving the text of the article. We have tried to take into account all of them.

  1. Regarding the Introduction: The introduction is extremely short and does not provide adequate background information. There are no clear explanations or references to previous biological or bioinformatics studies.

Thanks for this useful comment. We added more information about similar bioinformatic studies (lines 46-65). However, to our knowledge there are no articles describing pathways activated by an TRK fusion protein revealed by bioinformatic analysis. We also added it into the introduction section.

Regarding the MM section: In my opinion, this is the most problematic part of the manuscript. 

  1. First, I wonder why the authors use different pathway databases for their analysis, although according to the documentation of the -genexplain platform- all their analyses can be performed with only one of these databases. It needs to be explained why these different pathway databases were preferred and what is the difference between these pathway databases (Reactome, Transpath, Genways).

We were using three different pathway databases to obtain most reliable results. All three databases, TRANSPATH, Reactome and GeneWays are available in geneXplain platform. They differ significantly in the value of the information. GeneWays database was built on the basis of automatic text mining of scientific abstracts, so, the gene interaction network in this database contains a lot of indirect interactions and this database is used for generating broad hypothesis about potential upstream regulators of our genes. TRANSPATH database in turn is constructed on the basis of careful manual curation of mostly direct signal transduction reactions. It is used here to generate very precise hypothesis about most important key regulatory proteins. Combining results of analysis of our data using networks obtained from these two databases we on one side are guaranteed to get most reliable information and on the other side not to miss potential completely novel hypothesis about regulation of our genes. The Reactome database is used here as a very powerful collection of canonical pathways that complementary to the TRASPATH collection and it is very beneficial to use both databases for the analysis of pathway enrichment analysis. 

  1. According to the -genexplain platform- documentation, the underlying algorithm of "regulator search" and "upstream analysis" is the same. Both methods perform promoter analysis, find TFBSs and then identify master regulators.  It must be explained why the authors distinguish between them

We have added into the manuscript the following phrase: “Here the upstream analysis is empowered by the search for positive-feedback loops, that makes it more focused compared to the “Regulator search” (lines 104-106).

Again, the simple Regulator search allows to find all potential regulators of our genes, whereas the upstream analysis empowered by the search for positive feedback loops gives most significant key master-regulators. We use both approaches in order not to lose potentially interesting novel hypothesis and also ensure the most reliable results.

  1. Transfac database: which version and which PWM library with which cutoffs: this information is completely missing

The following sentences were added to the manuscript: “For the first step, the database TRANSFAC® [22] (release 2021.1 ) is employed together with the TF binding site identification algorithms Match [15], F-Match [16], and Compo-site Module Analyst (CMA) [17]. Match tool searches for TF binding sites in promoters of the input genes using the full collections of 7889 vertebrate position weight matrices (PWMs) from TRANSFAC database and applying PWM cut-offs “min_SUM” that provide the minimum of the sum of false positive and false negative rates in site prediction.” (lines 111-117).

  1. The authors must clearly mention that the -genexplain-platform-, the -genome-enhancer-platform- and the Transfac- and Transpath-databases are commercial. Therefore, the scripts provided in GitHub are only partially applicable. Otherwise, the link from GitHub is missunderstandable

We have included in the manuscript the following phrase: “In GitHub, the file Workflow_NTRK.dml provides the XML code of the workflow to ana-lyze a gene set and to predict master regulators. To execute the workflow, it should be up-loaded into geneXplain platform https://platform.genexplain.com. Licenses for the data-bases TRANSFAC, TRANSPATH, and HumanPSD are required to execute the full work-flow.” (lines 83-87).

  1. In general, the section on data analysis needs to be divided into several subsections. This is because, in my view, the authors should distinguish between master regulator analysis and gene set enrichment analysis. Also, it needs to be explained what a master regulator is with its underlying detection algorithm

We divided the Data analysis section of the Material and Methods chapter onto several subsections: Identification of differentially expressed genes (DEGs); Gene set enrichment analysis; and Master regulator analysis and upstream analysis (lines 79, 95, and 98).

Also, we included the phrase “Such potential rewiring in the network can be identified with the help of the search for positive feedback looks “walking pathways” [18] that is implemented in the Genome Enhancer tool.” in the section 3.4 to explain the rational of applying upstream analysis implemented in Genome Enhancer tool (lines 221-223).

  1. I wonder what the "Benjamini-Hochberg procedure for calculating false discovery rate" means. I know the Benjamini-Hochberg procedure used to control the false discovery rate. But I don't know how to calculate FDR based on BH procedure. Please explain it.

Thanks for the correction. We have changed the description in the manuscript: “the Bonferroni test or Benjamini-Hochberg procedure were used to control the false dis-covery rate” (line 127).

  1. Another problem is the use of the CMA method, which was published in 2006. First, the full name must be written before the abbreviation. But the authors need to explain why this approach is preferred in this study, although members of the genexplain platform have published new studies for the same purpose of "Composite Module Analysis". The studies are  "MatrixCatch" by A. Kel and E. Wingender (published 2013, Bmc Bioinformatics), and MatrixCatch is also integrated with genexplain. "PC-TraFF" and its update by E. Wingender (published 2015 Bmc Bioinformatics and 2018 Frontiers)

Thank you very much for the comment. We have added the explanation of the abbreviation CMA (Composite Module Analyst). The tools like MatrixCatch are used for searching for known composite elements. But CMA is a very important tool for contraction of yet unknown composite elements and more complex modules. Exactly such task was in our work here, that is why CMA was used. PC-TraFF algorithm indeed is very important as well, but it is limited only by pairs of TFs whereas CMA is looking for more complex combinations of TFs, that was important to search in this work in order to understand the full complexity of the processes observed.

  1. In this section, the explanation for the " walking pathways" must also be added.

We have mentioned in the text that the algorithm of “waking pathways” is looking for positive feedback loops and gave the reference to the paper where the algorithm was introduced (lines 221-223).

Regarding the results section: 

  1. In my opinion, the sentence in lines 118-119 is misleading: "...To select the most activated genes in response to NTRK fusion, we took genes with log2FoldChange≥1 ..." What does "most activated genes" mean: i) a very high number or ii) a very strong activation? If the authors wanted to study a very high number of genes, I could understand the use of a log2FC>=1, but for the most activated genes this threshold is not valid. This point needs to be clarified.

Thanks for this correction. We replaced it with “In order to select a high number of upregulated genes in response to NTRK-fusion, we took genes with log2FoldChange≥1 in any of these three comparisons and used them for further analysis” (lines 150-151).

  1. It is not clear to me how these numbers sum up in Figure 1 (B). Is there a total of 38391 down-regulated DEGs discovered by the authors, even though the Ensembl database currently provides information on 20465 vertebrate coding genes? This point needs to be explained.

The DEG analysis included all genes, coding, non-coding, and pseudogenes, the total number of which is about 60’000.

  1. In line 180, the authors mention enriched GO categories, which are listed in Supplementary Table 3. But none of these GO categories has an FDR value of less than 0.05. This threshold is defined by the authors for significance in section MM. It must be explained why these GO terms should be accepted as enriched even though their FDR values are not significant

We replaced it with the following phrase into the manuscript: “In total, we identified 29 upregulated genes that were related to the following GO categories associated with the nervous system (e.g., "axon," "nervous system process," "synapse assembly," "axonogenesis")” (line 209). In this part, the statistical significance is not so important because this GSEA was performed only to extract genes that are involved in the nervous system development like NTRK genes.

  1. In lines 192 to 194, the authors mention F-Match, an important tool that must be explained in the MM section. Further, the same lines say "... using the TRANSFAC motif library": which version of Transfac, which motif library. I think the authors are talking about a PWM library. If yes, it must be written clearly which PWM library with how many PWMs and with which threshold (minFN, minFP or minSUM or something else).

To explain the Match, F-Match and CMA tools we have introduced in the Method section (lines 111-123):

For the first step, the database TRANSFAC® [22] (release 2021.1 ) is employed together with the TF binding site identification algorithms Match [15], F-Match [16], and Compo-site Module Analyst (CMA) [17]. Match tool searches for TF binding sites in promoters of the input genes using the full collections of 7889 vertebrate position weight matrices (PWMs) from TRANSFAC database and applying PWM cut-offs “min_SUM” that provide the minimum of the sum of false positive and false negative rates in site prediction. F-Match tool identifies PWMs for transcription factors whose sites are statistically signifi-cantly enriched in the promoters of the input gene set in comparison with the background set of promoters (here a set of 1500 housekeeping genes is used as the background set). CMA tool searches for composite modules – combinations of the en-riched PWMs whose sites are co-localized in promoters of the studied genes and build compact clusters. Such composite modules often serve as functional units for regulation of gene expression in specific conditions (e.g. in the tumor cells).

To give more details about TRANSFAC we have introduced in the Method section:. “TRANSFAC® [22] (release 2021.1 )” (line 111).

  1. Regarding the discussion section: there is no relevant information about bioinformatics analysis. Why these databases and tools were selected, what are their advantages and disadvantages. The comparison or ignorance of other tools, as mentioned above. The discussion needs to be expanded in terms of bioinformatics analysis of the manuscript.

We have added the following paragraph in the Discussion section of the manuscript (lines 285-308):

In this work, in addition to the standard methods of pathway mapping and STRING analysis, we applied one of the most advanced approach of bioinformatics analysis of gene regulatory networks – upstream analysis and master regulator search implemented in the set of tools provided by geneXplain platform and Genome Enhancer. The upstream analysis approach tries to reconstruct the causal interactions in the signal transduction and gene regulatory networks that are heavily rewired in the tumor cells and cannot be properly mapped to canonical pathways that are acting in non-tumor cellular context. To do such causal reconstruction, the most comprehensive collection of TF binding site mo-tifs provided by TRANSFAC database was necessary to minimize the chances of missing an important TF-DNA interactions playing a vital role in the tumorigenic process. We also needed to understand which particular and rather novel combination of transcription factors is responsible for such very specific gene regulation that was observed in the stud-ied subtypes of tumors. The consequent use of the Match, F-Match and CMA tools pro-vided us this possibility, which is hardly possible with other tools, such as “MatrixCatch" [26], which searches only known composite elements but can not find new combinations, and "PC-TraFF"[27,28] which searches for TF site pairs in whole promoter length but not more complex combinations of co-localized TF sites. At the last step of the causal recon-struction of the rewired networks we searched for most probable regulators and key mas-ter-regulators. This was done with the use of GeneWays and TRANSPATH databases that gave us on one side a very broad scope of interactions (provided by the text mining based database GeneWays) and, on the other side, the highly accurate information about details of signal transduction reactions (provided by the manually curated and one of the most comprehensive database TRANSPATH). These combination of two databases helped us to achive the most reliable results of our analysis.

  1. Regarding the conclusion section: To meet the requirements for good academic work, a relevant conclusion is required. Please provide a proper conclusion

We added the following phrase into the conclusion: “Thus, using several modern bioinformatic approaches we managed to suggest new role for the ErbB4 in the pathway activation due to NTRK gene fusions in thyroid cancer.” (lines 369-370).

Reviewer 2 Report

Neurotrophic tyrosine receptor kinase (NTRK) gene fusions are present in many cancers, including in thyroid cancer, and they encode different TRK receptors that then are oncogenic drivers. However, the downstream pathways that these receptors activate are not completely understood. They are known to phosphorylate proteins that are then linked to the activation of certain oncogenes. The authors aimed to uncover more of those genes in thyroid cancer. To this end, they use a Bioinformatic approach and data from Cancer Genome Atlas data, DeSeq2, Genome Enhancer, and geneXplain platforms. In samples with NTRK fusion, they identified 29 genes related to nervous system development (e.g. AUTS2DTNAERBB4FLRT2FLRT3RPH3ASCN4A). In addition, upstream regulators of these genes were enriched in the "Signaling by ERBB4" pathway. They also found that ErbB4 is the potential key regulator of pathways activated by NTRK gene fusions in thyroid cancer. As mentioned by the authors, these results are preliminary and require additional biochemical validation. I suggest the following improvements:

- Abstract is unclear: It doesn´t allow a non-expert in these pathways to understand easily the article. 

- Rest of the text: also need clarifications to make it clear for non-experts in this particular subject.

- Figure 1, 3,4,5, 7: needs better resolution;

- In figure 2, the lettering is too small and the grey color makes it difficult to see.

Below are some sentences with corrections in CAPITAL letters:

- “The data preprocessing, normalization, identification of differentially expressed genes (DEGs), and visualization WAS”

- NTRK fusions were detected with ETV6-NTRK3 specific primers and probe, ERBB4 and TBP expression WERE analyzed

- The p-values were not used for gene filtering due to the low number of samples with NTRK-fusion WHICH could lead to...

Author Response

  1. Abstract is unclear: It doesn´t allow a non-expert in these pathways to understand easily the article.

We added several explanations into the Abstract: “We found that genes regulating expression of the upregulated genes (i.e. upstream regulators) were enriched in "Signaling by ERBB4" pathway.” and “Moreover, the algorithm searching for positive feedback loops for gene promoters and transcription factors (so called "walking pathways" algorithm) identified the ErbB4 protein as the key master regulator”. (lines 24-28).

  1. Rest of the text: also need clarifications to make it clear for non-experts in this particular subject.

We added more information about the algorithms used into the Material and Methods section.

  1. Figure 1, 3,4,5, 7: needs better resolution;

We replaced figures 1, 3, 4, and 5 with high-resolution figures. However, Figure 7 seems to be already high-resolution.

  1. In figure 2, the lettering is too small and the grey color makes it difficult to see.

We uploaded the figure with higher resolution but unfortunately colors cannot be changed. Nevertheless, this figure reflects the enrichment of nervous system processes by the upregulated genes identified.

  1. Below are some sentences with corrections in CAPITAL letters:

- “The data preprocessing, normalization, identification of differentially expressed genes (DEGs), and visualization WAS”

- NTRK fusions were detected with ETV6-NTRK3 specific primers and probe, ERBB4 and TBP expression WERE analyzed

- The p-values were not used for gene filtering due to the low number of samples with NTRK-fusion WHICH could lead to...

Thanks for these corrections. We have introduced them into the manuscript.

Round 2

Reviewer 1 Report

The authors have been very responsive to all my comments, and the manuscript meets the requirements of a good scientific paper,

From my point of view it is now acceptable. 

Congratulations!